# Atmospheric nitrogen deposition fluxes into coastal wetlands and their impacts on ecosystem carbon sequestration in East Asia

Jia Liu[1], Yan Zhang[1,2,3], Shenglan Jiang[1], Yuqi Xiong[1], Chenji Jin[1], Qi Yu[1], Weichun Ma[1]

[1]Shanghai Key Laboratory of Atmospheric Particle Pollution and Prevention, National Observations and Research Station for Wetland Ecosystems of the Yangtze Estuary, Department of Environmental Science and Engineering, Fudan University, Shanghai 200433, China

[2]MOE laboratory for National Development and Intelligent Governance, Shanghai institute for energy and carbon neutrality strategy, IRDR ICoE on Risk Interconnectivity and Governance on Weather/Climate Extremes Impact and Public Health, Fudan University, Shanghai 200433, China

[3] Shanghai Institute of Eco-Chongming (SIEC), Shanghai, 200062, China

*Correspondence to*: Yan Zhang (yan_zhang@fudan.edu.cn)

**Abstract.** Coastal wetlands serve as critical sinks for both carbon and nitrogen within regional ecosystems, playing an essential role in mitigating atmospheric greenhouse gases and nutrient enrichment. This study integrates high-resolution wetland type data, AIS-based ship emission inventories, and regional nitrogen deposition simulations to quantify nitrogen inputs to East Asian coastal wetlands from the perspective of source–sink coupling. Firstly, the atmospheric nitrogen deposition fluxes in coastal wetland areas of East Asia were simulated and evaluated with WRF-CMAQ. Nitrogen deposition fluxes were spatially coupled with classified wetland maps in ArcGIS. Net primary productivity (NPP) was estimated using a modified CASA light-use efficiency model, incorporating solar radiation and FPAR from remote sensing. Carbon sequestration and oxygen release were then quantified using stoichiometric relationships based on NPP. The results indicate that total nitrogen deposition across East Asian coastal wetlands follows a general gradient of "high in the south, low in the north" and "strong in urban-industrial clusters, weak in remote coastal zones." On average, ship emissions contribute 10.13 % and 15.22 % to $NO_3^-$-N and $NH_4^+$-N deposition, respectively, while their contribution to gaseous $NH_3$-N is negligible. Among wetland types, salt marshes receive the highest nitrogen input per unit area (654.99 mg $NO_3^-$-N·m$^{-2}$·yr$^{-1}$), although tidal flats dominate total regional nitrogen input due to their extensive spatial coverage. Dry and wet deposition exhibit significant seasonal variation: wet deposition consistently prevails during the spring and summer months due to frequent rainfall, while dry deposition becomes increasingly prominent in autumn and winter. For instance, in the Korean Peninsula, the wet-dry gap in nitrate deposition reaches 0.17 g N·m$^{-2}$·yr$^{-1}$, while the Yangtze River Delta exhibits relatively balanced ammonium inputs (dry-wet difference of only 0.05 g N·m$^{-2}$·yr$^{-1}$). Carbon sequestration capacity shows strong spatial and temporal coupling with nitrogen deposition. Mangrove forests exhibit the highest annual NPP (~776.16 g C·m$^{-2}$·yr$^{-1}$ in summer), supported by high FPAR and solar radiation (1749.29 MJ·m$^{-2}$), followed by salt marshes and tidal flats. Seasonal patterns reveal a summer peak in carbon uptake across all wetland types, with mangrove NPP in summer being two times higher than winter values. Nitrogen deposition primarily enhances carbon sequestration during warm seasons; for instance, in the mangroves of the Pearl River Delta, nitrogen inputs increase summer $CO_2$ fixation by 6.85 g C·m$^{-2}$, while the effect is negligible in winter (<0.06 %) or in nitrogen-saturated regions. These findings provide a scientific foundation for understanding how coastal ecosystems respond to anthropogenic activities and long-range nitrogen transport. Furthermore, the results serve as an important reference for wetland conservation, nitrogen cycle management, and the development of regional carbon neutrality strategies.

## 1 Introduction

Coastal wetlands, functioning as dynamic interfaces between terrestrial and marine systems, serve not only as critical habitats for global biodiversity but also as integral components of regional and global ecosystems (Wang et al., 2025; Huang et al., 2025). In East Asia, these wetlands exhibit considerable heterogeneity, encompassing mangrove forests, salt marshes, mudflats, coral reefs, and seagrass beds—each characterized by distinct ecological functions. Mangrove forests mitigate storm surges through their complex root systems and play a vital role in carbon sequestration. Globally mangrove habitats stocked an

average of 693 t $C_{org}$ ha$^{-1}$, which is 15–24 % of the tropical coastal ocean carbon burial rate of 184 g $C_{org}$ m$^{-2}$ yr$^{-1}$ (Rani et al., 2023). Salt marshes, with salt-tolerant vegetation, can retain heavy metals and organic pollutants in response to tidal dynamics. Mudflats mitigate flood and tidal impacts by facilitating sediment deposition. Coral reefs and seagrass beds sustain marine biodiversity through biocalcification and primary production. In recent decades, however, coastal wetlands have faced increasing threats from land reclamation, industrial and agricultural pollution, and climate change. Consequently, their areal extent has declined by approximately 35 %, according to regional remote sensing data, and the degradation of their ecological functions poses serious challenges to the global carbon cycle and coastal zone stability (Barbier et al., 2011; Goldberg et al., 2020; Li et al., 2018b).

Against this backdrop, nitrogen deposition, resulting from anthropogenic emissions and atmospheric chemical processes (Gruber and Galloway, 2008), has emerged as a key driver reshaping the ecological dynamics of coastal wetlands. Acting as a "double-edged sword," atmospheric nitrogen—delivered via dry and wet deposition—directly influences NPP and carbon sequestration potential. Nitrogen and carbon cycles are inherently interconnected (Cheng et al., 2018). Previous studies have shown that moderate nitrogen inputs (<10 kg N ha$^{-1}$ yr$^{-1}$ can stimulate photosynthetic activity in mangrove and salt marsh vegetation, enhance biomass accumulation, and promote the storage of soil organic carbon (Rong and Rui-Ying, 2015; Adam Langley et al., 2013; Morris and Bradley, 1999). However, rapid industrialization and agricultural intensification across East Asia have led to nitrogen deposition levels exceeding 20 kg N ha$^{-1}$ yr$^{-1}$ in some regions, such as the Yangtze River Delta (YRD). This excessive nitrogen may induce an imbalance in the nitrogen-to-phosphorus ratio, potentially triggering cascading ecological effects (Liu et al., 2013). On one hand, excessive nitrogen promotes microbial heterotrophic respiration, accelerating the mineralization of soil organic matter and potentially transforming wetlands from carbon sinks into carbon sources. On the other hand, intensified nitrification–denitrification processes may increase the emission of greenhouse gases such as $N_2O$ and disrupt symbiotic relationships between benthic fauna and plant communities, potentially leading to biodiversity loss. Moreover, the ecological impacts of nitrogen deposition are spatially heterogeneous: mangrove forests may retain nitrogen due to their anaerobic root environments, while unvegetated mudflats may facilitate nitrogen migration via surface runoff, increasing the risk of eutrophication in adjacent marine waters (Sasmito et al., 2020). Therefore, quantifying nitrogen deposition fluxes and their spatiotemporal distribution is critical for understanding the atmospheric nitrogen inputs into coastal wetland ecosystems. A comprehensive understanding of both the intensity and variability of nitrogen input is essential for enhancing carbon sequestration, optimizing nutrient regulation, and supporting the achievement of regional carbon neutrality targets.

Coastal wetlands, in addition to their role as biodiversity hotspots, also function as critical regulators of the carbon–nitrogen–phosphorus (C–N–P) cycles. However, multiple anthropogenic and climatic stressors—such as land reclamation, industrial and agricultural emissions, and sea-level rise—have led to a 40 % reduction in the total area of coastal wetlands in East Asia in recent decades (Yoshikai et al., 2021; Li et al., 2020; Leidner and Buchanan, 2018), substantially weakening their ecological functions and threatening the stability of the global carbon cycle and coastal systems.

In the international context, nitrogen deposition has become a focal issue in global change ecology, particularly concerning its ecological impacts on coastal wetlands. Long-term studies in Europe and North America have explored nitrogen input mechanisms across various wetland types. For instance, observations in the salt marshes of Chesapeake Bay, USA, have demonstrated that atmospheric nitrogen deposition can significantly alter the soil nitrogen-to-phosphorus ratio (N:P > 16), thereby suppressing reed root development and reducing wetland carbon sequestration efficiency by 12 %–18 % (Deegan et al., 2012; Morris, 1991). Research along the North Sea coast further shows that nitrogen deposition, in conjunction with tidal hydrodynamics, accelerates the mineralization of sedimentary organic nitrogen and enhances $N_2O$ emissions via nitrification, contributing approximately 7 %–9 % to regional greenhouse gas budgets (Marchant et al., 2018; Barnes and Upstill-Goddard, 2011; Bange, 2006). In tropical mangrove systems, studies from Southeast Asia reveal that excessive nitrogen deposition (>25 kg N ha$^{-1}$ yr$^{-1}$) inhibits symbiotic nitrogen-fixing bacteria and facilitates the invasion of non-native species such as Spartina alterniflora, ultimately destabilizing native ecosystems (Xia et al., 2021; Chakraborty, 2019; Porter et al., 2013). However, these studies often focus on individual wetland types within specific climate zones and lack integrative, cross-scale assessments of nitrogen deposition dynamics in the ecologically diverse coastal wetlands of East Asia.

Recent domestic research has advanced our understanding of this issue. Field observations in the salt marshes of the YRD

indicate that ammonium nitrogen ($NH_4^+$) from industrial emissions accounts for approximately 65 % of total atmospheric nitrogen deposition in the region(Xu et al., 2018). While such nitrogen input temporarily enhances carbon sequestration by promoting the biomass of Suaeda salsa (Hong et al., 2024; Chen and Sun, 2020), long-term deposition leads to soil acidification (pH decline of 0.3–0.5) and increased mobility of heavy metals, such as a 40 % rise in cadmium activity (Hu et al., 2019). Modelling studies by Wu et al. further show that during the first two decades of elevated nitrogen deposition, plant productivity, autotrophic respiration, and litter production increase sharply, resulting in significant alterations to ecosystem carbon fluxes. Beyond this initial phase, carbon cycling becomes constrained primarily by vegetation composition and biomass distribution within the newly stabilized ecosystem (Wu et al., 2015). In the mudflats of Bohai Bay, where vegetation cover is minimal, studies suggest that 30 %–50 % of deposited nitrogen is transported into adjacent marine environments via surface runoff, contributing significantly to the eutrophication of the Yellow Sea (Fu et al., 2021; Zheng and Zhai, 2021). Despite these insights, existing research remains largely limited to single environmental media, specific wetland types, or short temporal scales. There is a pressing need for systematic, regionally comparative assessments of nitrogen deposition patterns and their ecological consequences across multiple wetland categories in East Asia.

In view of the significant disparities that exist among various coastal wetland types, their carbon and nitrogen fixation pathways, efficiencies, and ecosystem responses are influenced by multiple factors, including vegetation characteristics, soil properties, hydrodynamics, and nitrogen input intensity (Fettrow et al., 2024; Alongi, 2020). Given the dual roles of coastal wetlands as carbon and nitrogen sinks, their functions have attracted increasing attention in the context of climate change mitigation and regional environmental governance. Nevertheless, a comprehensive understanding of nitrogen deposition fluxes, spatiotemporal variability, and their regulatory influence on carbon sequestration processes remains incomplete. To address this gap, the present study focuses on six representative coastal wetland systems in East Asia—specifically mangrove forests, salt marshes, and tidal flats. The objectives are to (1) examine the seasonal and regional variation of nitrogen deposition fluxes under different climatic conditions and vegetation types; (2) quantify their impacts on NPP and $CO_2$ fixation; and (3) analyse the coupling mechanisms between nitrogen deposition and carbon sink capacity. This research aims to elucidate key ecological processes and spatial heterogeneity in the synergistic regulation of carbon and nitrogen cycles in coastal wetlands. The findings are expected to support improved assessments of ecosystem services, inform nitrogen pollution control policies, and contribute to wetland conservation strategies aligned with regional carbon neutrality goals.

## 2 Materials and Methods

The East Asia region is located in the eastern part of Asia, bordering the Pacific Ocean to the east. It encompasses countries and regions such as China, Japan, South Korea, and North Korea, featuring diverse geographical environments and rich ecosystems. This region is not only one of the most dynamic areas in global economic development, but also boasts a coastal wetland system of great ecological value. Coastal wetlands in East Asia are widely distributed along the eastern coast of China, the west coast of the Korean Peninsula (KP) and the coast of the Japanese archipelago. They mainly include types such as salt marshes, mudflats, mangrove forests and estuarine deltas. These wetlands play an irreplaceable ecological role in maintaining biodiversity, resisting storm surges, and purifying water quality.

### 2.1 Emission and Land Cover in East Asia

This study focuses on the East Asian region, including mainland China and neighbouring countries. For mainland China, the 2017 Multi-resolution Emission Inventory for China (MEIC) (Zheng et al., 2018; Li et al., 2017a) was used as the anthropogenic emissions dataset. For other regions of East Asia, the 2010 MIX Asian emissions inventory was employed (Li et al., 2017b). The ship-based emission inventory for conventional atmospheric pollutants was constructed using a bottom-up approach based on real-time data from the 2017 Automatic Identification System (AIS). This method calculates emissions using fuel consumption-based emission factors specific to different marine fuels and adjusts for individual vessel characteristics, such as engine type and operational status (Yuan et al., 2023; Fan et al., 2016). The resulting ship emissions inventory includes nitrogen oxides ($NO_x$), ammonia ($NH_3$), and particulate matter ($PM_{2.5}$, $PM_{10}$), sulphur dioxide ($SO_2$), $NO_x$, carbon monoxide (CO), hydrocarbons, and greenhouse gas (GHG) species (Yi et al., 2025). Detailed computational procedures

are provided in Supplementary Section 1.1. Emission factors used for these species are listed in Supplementary Table S1, with units expressed in grams per kilowatt-hour (g (kWh)$^{-1}$).

AI Earth (https://engine-aiearth.aliyun.com) is a cloud-based geospatial platform that provides a wide range of environmental datasets, including meteorological, land surface, and atmospheric composition variables. In this study, datasets were selected based on their temporal coverage, spatial resolution, and relevance to the research objectives. Specifically, the 2017 Global 30-meter Wetland Dataset (GWL_FCS30) (Zhang et al., 2024) was utilized. Using AI Earth's integrated data processing tools, the dataset was filtered to extract data corresponding to the required time periods, geographic regions, and variable subsets.

These preprocessing options ensured that only data pertinent to the study area were retained. The dataset was then clipped to the predefined study region to reduce data volume and ensure spatial compatibility. Where necessary, advanced tools were employed to perform coordinate system transformations. Overall, the nitrogen (N element) emission inventory and wetland type distribution in East Asia adopted in this study are shown in **Fig. 1**.

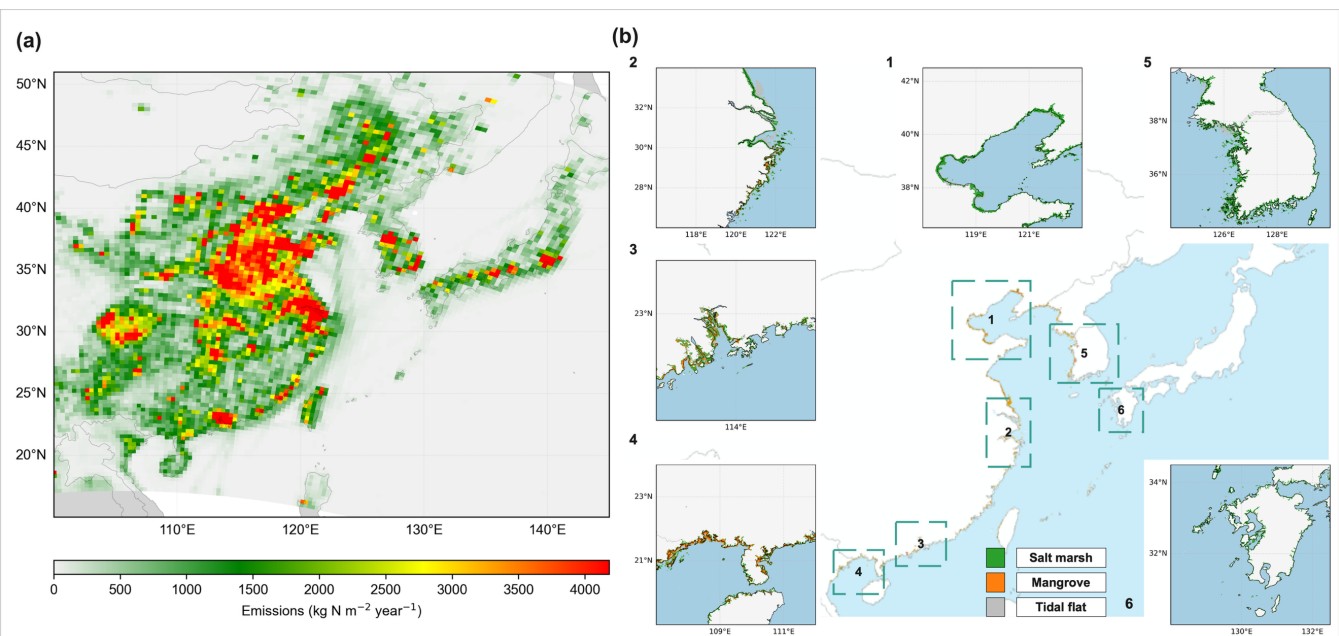

**Figure 1. Nitrogen emissions (a) and wetland type distribution (b) in East Asia. Different research areas in East Asia have been marked with dotted lines.**

## 2.2 Atmospheric Deposition Model System Description

This study employed a mesoscale meteorological–chemical modelling framework to simulate the transport and deposition of atmospheric nitrogen in coastal regions. Meteorological fields were generated using the Weather Research and Forecasting (WRF) model version 4.1.1, driven by global atmospheric reanalysis data from the National Centers for Environmental Prediction (NCEP) Final Operational Global Analysis (FNL)(Commerce, 2000), which serves as both the initial and boundary condition dataset. The input data have a spatial resolution of 1° × 1° and a temporal resolution of 6 hours. CMAQ version 5.4

was used to estimate nitrogen deposition in both oxidized and reduced forms. The simulation domain covered the East Asian region, as illustrated in Fig. S1, with detailed parameter settings provided in Supplementary Section 1.2.

In developing the emission inventories, this study explicitly separated terrestrial anthropogenic sources from marine shipborne emissions to enable a source-specific attribution of atmospheric nitrogen inputs to coastal wetlands. The land-based anthropogenic emissions for China were derived from the Multi-resolution Emission Inventory for China (MEIC), while

emissions for other Asian regions were based on the MIX Asian inventory(Yue et al., 2017). Shipborne emissions were calculated using a bottom-up method based on AIS data, following established practices for high-resolution marine emission modelling (Jiang et al., 2024; Fan et al., 2016). Based on the baseline land-based anthropogenic inventory, two parallel emission scenarios were constructed. The first scenario included both land and marine emissions, hereafter referred to as the

with-shipping scenario. The second scenario excluded all shipborne emissions, hereafter referred to as the without-shipping scenario. The contribution of shipping to nitrogen deposition was quantified by comparing the deposition fields simulated under these two scenarios.

In this study, total nitrogen (TN) refers exclusively to total inorganic nitrogen (TIN), defined as the sum of oxidized inorganic nitrogen species ($NO_2$, $NO$, $NO_3^-$) and reduced inorganic nitrogen species ($NH_3$ and $NH_4^+$). TIN was simulated for four representative months of 2017 (January, April, July and October), corresponding to winter, spring, summer and autumn. Selecting single representative months has been widely adopted in regional modelling to capture climatological seasonal characteristics under factorial experimental designs (Li et al., 2019; Qi et al., 2017). Each simulation used a five-day spin-up period to minimize the influence of initial conditions. In total, eight model experiments were conducted, consisting of four months and two emission scenarios. This simulation framework provided a consistent basis for evaluating both seasonal variations and the source-specific contributions of nitrogen deposition in East Asian coastal wetlands.

The use of a single representative month for each season is a methodological simplification relative to full three-month seasonal simulations. This choice was dictated by the factorial experimental design, which required independent simulations under two emission scenarios, and by the associated computational demands. Although this single-month representation is widely adopted in regional atmospheric modelling studies and has been demonstrated to capture the characteristic meteorological and chemical features of each season(Wu et al., 2021; Li et al., 2018a), it inevitably introduces some degree of uncertainty related to intra-seasonal variability. Future work involving continuous multi-month simulations for each season would help further constrain this uncertainty.

## 2.3 Calculation of Carbon Sequestration

Coastal wetlands in East Asia play a significant role in carbon sequestration. According to previous studies, 1 kg of carbon corresponds to approximately 2.2 kg of organic matter. Based on the chemical equations of photosynthesis, the production of 1 kg of organic matter leads to the fixation of approximately 1.63 kg of $CO_2$ and the release of 1.19 kg of $O_2$. Additionally, 1 kg of $CO_2$ contains about 0.27 kg of elemental carbon (Wang et al., 2005; Minghui et al., 2012). Using these stoichiometric relationships, annual carbon sequestration and oxygen release from wetland ecosystems can be estimated based on their net primary productivity (NPP).

The following equations were used to calculate carbon sequestration and oxygen release:

$$W_{CO2} = NPP \times 2.2 \times 1.63 \ , \tag{1}$$

$$W_C = W_{CO2} \times 0.27 \ , \tag{2}$$

$$W_O = NPP \times 2.2 \times 1.19 \ , \tag{3}$$

where $W_{CO2}$ represents the amount of $CO_2$ fixed per unit area (g m$^{-2}$), Wc represents the carbon sequestration per unit area of wetland (g m$^{-2}$), NPP refers to the net primary productivity (g C m$^{-2}$) of vegetation per unit area per year, and Wo represents the oxygen release per unit area of wetland (g m$^{-2}$).

NPP was estimated using a modified Carnegie–Ames–Stanford Approach (CASA) model (Tang et al., 2024; Su et al., 2021), which simulates annual and seasonal NPP values based on remote sensing imagery and meteorological data. Seasonal values were extracted for January, April, July, and October to represent winter, spring, summer, and autumn, respectively. The CASA model, a light use efficiency model, integrates both environmental and biological factors—including effective solar radiation and vegetation-specific characteristics—to estimate NPP.

The basic structure of the CASA model is as follows:

$$NPP = APAR \times \varepsilon \ , \tag{4}$$

$$APAR = SOL \times FPAR \times 0.5 \ , \tag{5}$$

where absorb photosynthetically active radiation (APAR) represents the photosynthetically active radiation absorbed by the vegetation (MJ m$^{-2}$), $\varepsilon$ is the light use efficiency (g C MJ$^{-1}$), SOLAR refers to the total monthly solar radiation (MJ/ m$^2$), FPAR

is the fraction of photosynthetically active radiation (FPAR) absorbed by the vegetation canopy, and the constant 0.5 represents the proportion of incoming solar radiation that is photosynthetically active.

In the CASA model, biome-specific constant FPAR values were assigned to different coastal wetland types to reflect their contrasting canopy structures and vegetation cover. Specifically, an FPAR of 0.85 was used for mangroves, consistent with satellite-derived APAR estimates for dense mangrove forests (Zheng and Takeuchi, 2022). A moderate FPAR of 0.65 was adopted for salt-marsh wetlands, in line with typical growing-season FPAR ($\approx$0.4–0.7) reported for marsh vegetation. For sparsely vegetated tidal flats, an FPAR value of 0.10 was chosen to represent the dominance of water and bare sediment and the low emergent leaf area during most tidal cycles (Hawman et al., 2023). The light use efficiency $\varepsilon$ was parameterized using values derived from existing literature. For forests, a value of 1.044 g C MJ$^{-1}$ was used, while for salt marshes, a lower bound of 0.608 g C MJ$^{-1}$ was adopted (Zhu et al., 2006). Monthly solar radiation data used in the model are listed in Supplementary Table S3.

## 3 Results and Discussion

### 3.1 Nitrogen Deposition Flux in East Asian Coastal Wetlands

The nitrogen deposition flux in East Asian coastal wetlands exhibits pronounced source-specific heterogeneity. In terms of overall magnitude, terrestrial anthropogenic activities remain the dominant contributors to atmospheric nitrogen inputs, substantially exceeding the influence of shipborne emissions (Table 1), consistent with previous estimates of East Asian reactive nitrogen budgets (Liu et al., 2013). Nevertheless, marine ships exert a non-negligible impact on specific chemical forms and in particular coastal zones, especially in regions with dense where ship emissions significantly enhance coastal NO$_x$ and particulate nitrogen levels (Johansson et al., 2017; Lv et al., 2018).

Table 1. The totally different source of nitrogen deposition in different wetlands (Unit: mg N m$^{-2}$ year$^{-1}$)

|  | Sources | Salt marsh | Mangrove forest | Tidal flat | All |
|---|---|---|---|---|---|
| NO$_3^-$-N | Anthropogenic | 588.97 | 516.69 | 534.99 | 540.11 |
|  | Marine ship | 66.02 | 43.07 | 60.99 | 60.86 |
|  | Total | 654.99 | 559.76 | 595.98 | 600.97 |
| NH$_4^+$-N | Anthropogenic | 112.20 | 118.86 | 93.17 | 96.23 |
|  | Marine ship | 18.94 | 16.16 | 17.11 | 17.27 |
|  | Total | 131.14 | 135.02 | 110.28 | 113.50 |
| NO$_x$-N | Anthropogenic | 77.01 | 62.21 | 73.19 | 73.20 |
|  | Marine ship | 9.40 | 7.08 | 10.13 | 9.93 |
|  | Total | 86.41 | 69.29 | 83.32 | 83.13 |
| NH$_3$-N | Anthropogenic | 59.96 | 72.36 | 55.54 | 56.67 |
|  | Marine ship | 0.01 | 0.01 | 0.01 | 0.01 |
|  | Total | 59.98 | 72.38 | 55.55 | 56.68 |

Note: The last column of the table represents the nitrogen deposition flux for each wetland type, which is the weighted average value calculated based on the area proportion of each wetland.

According to simulation results, nitrogen deposition fluxes from ship emissions were 60.86 mg N m$^{-2}$ yr$^{-1}$ for NO$_3^-$-N, 17.27 for NH$_4^+$-N, 9.93 for NO$_x$-N, and 0.01 for NH$_3$-N, respectively. These values correspond to contribution rates of 10.13 %, 15.22 %, 11.95 %, and 0.02 %. Although the contribution of ships to gaseous NH$_3$-N is negligible, they accounted for over 10% of the total particulate nitrogen deposition (NO$_3^-$-N and NH$_4^+$-N). This highlights the role of ship emissions in the regional

formation of nitrate aerosols and secondary particulate matter, especially in coastal atmospheric environments where both dry and wet deposition processes are prevalent (Chen et al., 2024; Qiu et al., 2022). Deposition patterns also differ significantly across various coastal wetland types. A comparative analysis of three representative wetlands, including mangrove forests, salt marshes, and tidal flats, reveals marked differences in gaseous nitrogen deposition. The total deposition of $NO_3^-$_N and $NH_4^+$_N in mangrove forests wetlands was 559.76 mg N m$^{-2}$ yr$^{-1}$ and 135.02 mg N m$^{-2}$ yr$^{-1}$, respectively, which is lower than the deposition in salt marsh wetlands (654.99 and 131.14 mg N m$^{-2}$ yr$^{-1}$). In contrast, the deposition flux of $NH_3$-N was observed to be 17.13 % and 23.25 % lower in salt marshes and tidal flats, respectively, when compared to mangrove forests. These discrepancies are likely attributed to variations in wetland distribution, microclimatic conditions, and vegetation structure. Mangrove ecosystems, typically situated in estuarine regions, are characterized by complex vegetation that enhances the capture of both particulate nitrogen and reactive gaseous nitrogen (Alongi, 2020). Tidal flats often function as atmospheric nitrogen sinks due to their proximity to riverine and marine interfaces (Wang et al., 2019). This insight aligns with global analyses from NASA's nitrogen deposition synthesis, which emphasize the necessity of integrating land cover extent into regional nitrogen cycle assessments (Vet et al., 2014).

In summary, nitrogen deposition in East Asian coastal wetlands is shaped jointly by the type of emission sources and the chemical forms of nitrogen involved. Terrestrial anthropogenic emissions remain the primary contributor to total reactive nitrogen inputs, which is consistent with continental-scale assessments across China and other industrialized regions (Liu et al., 2024, 2013). However, shipborne emissions also exert an important influence on coastal atmospheric chemistry, particularly through the formation of secondary particulate nitrate and ammonium. The simulated contributions of shipping in this study align with previous findings that reported substantial maritime impacts on coastal nitrogen aerosols. Evidence from regional atmospheric modelling consistently indicates that shipborne $NO_x$ emissions substantially elevate coastal nitrate concentrations, with increases of 20–30% reported for Chinese marginal seas (Lv et al., 2018), and analogous nitrate enhancements found in the Yangtze River Delta due to dense maritime traffic (Liu et al., 2016). Compared with these studies, our results reveal a comparable magnitude of ship-related nitrogen deposition, particularly for particulate $NO_3^-$–N and $NH_4^+$– N in areas with intensive port activity. This consistency reinforces the emerging understanding that maritime transport represents a significant and spatially focused source of nitrogen enrichment in coastal boundary layers, complementing the broader and more diffuse contributions from land-based anthropogenic sources.

## 3.2 Spatial Temporal Differences in Nitrogen Deposition Fluxes of Different Coastal Wetlands in East Asia

### 3.2.1 Spatial Characteristics of Nitrogen Deposition in East Asian Coastal Wetlands

Spatial distribution maps of nitrogen deposition across six coastal wetland regions in East Asia revealed a general gradient characterized by "higher in the west, lower in the east" and "stronger in the south, weaker in the north" (**Fig. 2**). The distribution of nitrogen deposition points across countries is further illustrated in Supplementary Fig. S1. Among the six regions, the YRD, Pearl River Delta (PRD), and Beibu Gulf (BG) exhibited the highest nitrogen deposition fluxes, with localized values exceeding 10 g N·m$^{-2}$ yr$^{-1}$, highlighting substantial anthropogenic influence in these areas. In the Bohai Bay (BS), although the overall nitrogen deposition intensity is slightly lower than that in southern regions, industrial clusters near cities such as Tianjin and Tangshan show distinctly elevated fluxes. Wetlands on the western KP and Japan's Kyushu Island had scattered patterns of low to moderate nitrogen deposition that were largely influenced by atmospheric transport and local topography.

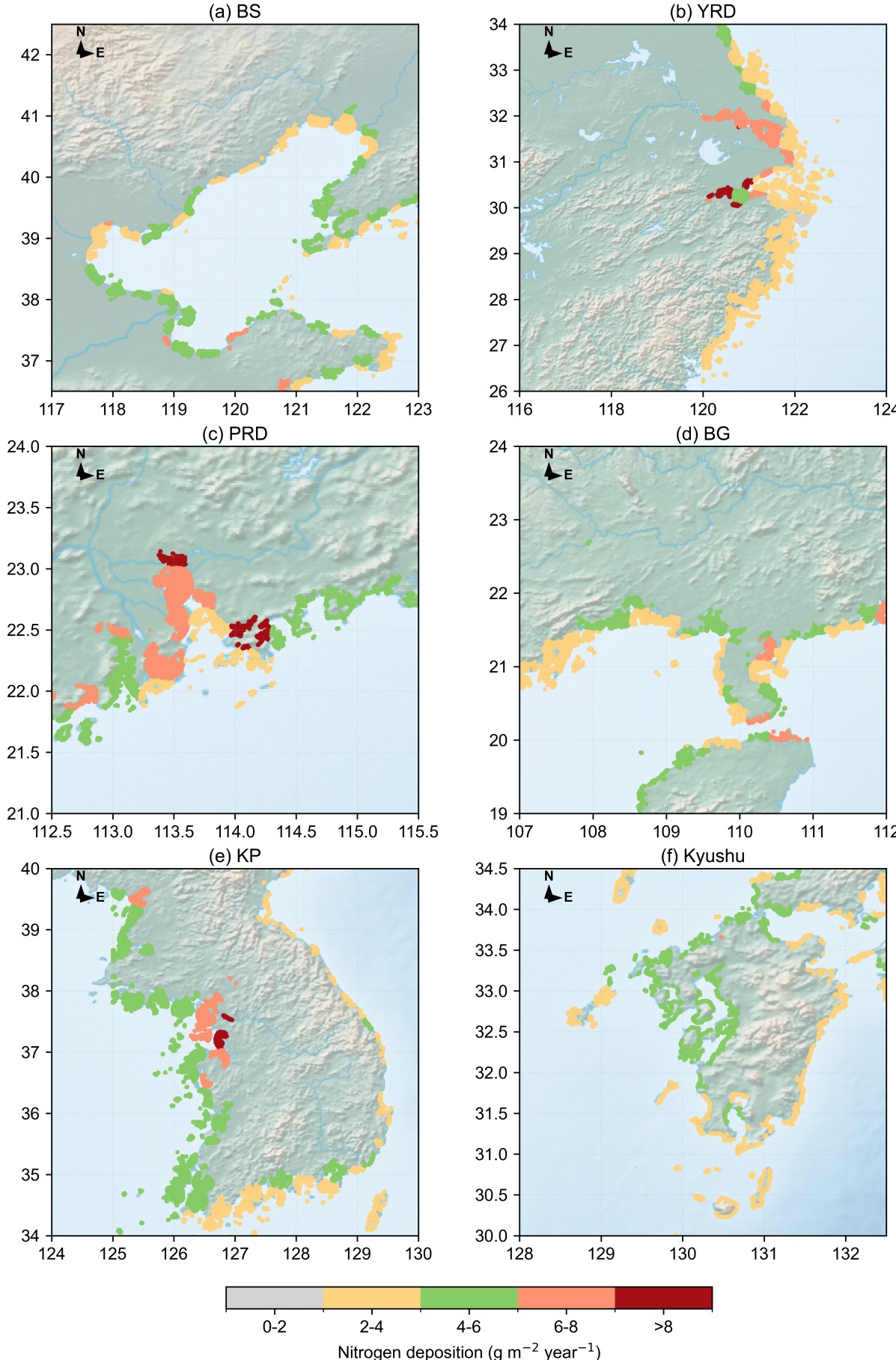

**Figure 2. Spatial Distribution of annual total atmospheric nitrogen deposition in coastal wetlands across different regions of East Asia (g N m⁻² yr⁻¹). BS means Bohai Bay, YRD means the Yangtze River Delta, PRD means Pearl River Delta, BG means Beibu Gulf, KP means the Korean Peninsula, and Kyushu means the Kyushu of Japan.**

Moderate nitrogen deposition is typical for the wetlands of BS, but some areas, especially near the Beijing-Tianjin-Hebei urban agglomeration, could exceed 8 g N m$^{-2}$ yr$^{-1}$. Fluctuations in wetlands on the Shandong Peninsula, especially in the Bohai Bay area, are significantly higher than those in other parts of the bay. In the YRD, nitrogen deposition rates typically ranged from 2 to 4 g N m$^{-2}$ yr$^{-1}$, with the highest values predominantly observed in riverine and coastal zones such as the Yangtze River estuary, Hangzhou Bay, and the Zhejiang coastal region. These elevated deposition levels were likely attributable to the dense distribution of emission sources in industrialized and urbanized areas (Liu et al., 2013). Similarly, wetlands in the PRD exhibited considerable nitrogen deposition, particularly in the Pearl River Estuary, where concentrations increased with proximity to an inland urban centre. This spatial pattern reflects the combined influence of industrial activities and maritime shipping. The ten key strategic pillar industries of Guangdong Province were largely concentrated in this core region (Huang et al., 2015), and major ports such as Guangzhou, Shenzhen, and Zhuhai constitute a globally significant maritime hub. Consequently, ship emissions contributed significantly to the observed nitrogen deposition. The wetlands demonstrated a distinct spatial pattern characterized as "combined land-sea source influence with regional retention effects." In cities including Nanning, Fangchenggang, Qinzhou, and Beihai, average nitrogen deposition was approximately 0.69 g N m$^{-2}$ yr$^{-1}$, with peak values ranging between 6 and 8 g N m$^{-2}$ yr$^{-1}$. In contrast, coastal wetlands on the KP generally experience lower nitrogen input levels, although higher deposition rates are observed on certain offshore islands, likely due to localized accumulation and long-range atmospheric transport (Jo et al., 2025). On a regional scale, nitrogen deposition in the KP showed a tendency to accumulate in northern and inland areas, potentially influenced by transboundary transport from China and local emission sources. On Kyushu Island in Japan, nitrogen deposition was primarily in the form of NO$_3^-$-N, with an average flux of approximately 0.14 g N m$^{-2}$ yr$^{-1}$. Elevated deposition levels were mainly found along the north-western coastline and in the northern regions of the island, likely driven by atmospheric transport from coastal Chinese regions and anthropogenic activities on adjacent Japanese islands (Morino et al., 2011).

In summary, nitrogen deposition across East Asian coastal wetlands exhibited marked spatial heterogeneity driven by a combination of land-based emissions, maritime shipping lanes, atmospheric transport, and geographic/topographic factors. Coastal regions such as the YRD and PRD, which are economically dynamic, experience the highest levels of nitrogen input. Meanwhile, the BS and BG display more uneven and spatially fragmented deposition patterns. Wetlands in the KP and Kyushu Island, while exhibiting lower overall fluxes, reflected the influence of long-range nitrogen transport and localized accumulation. These findings emphasize the need for region-specific nitrogen management strategies that account for both emission source profiles and spatial variability in ecosystem sensitivity.

**3.2.2 Dry and Wet Deposition Fluxes and Seasonal Characteristics of Nitrogen in Coastal Wetlands of East Asia**

Nitrogen deposition in coastal wetlands primarily occurs through two mechanisms: wet and dry deposition (Seinfeld and Pandis, 2016). Wet deposition involves the removal of atmospheric nitrogen compounds, both gaseous and particulate, via precipitation (such as rain or snow), occurring mainly through in-cloud (rainout) and below-cloud scavenging processes (Xu et al., 2020). In contrast, dry deposition refers to the direct transfer of nitrogen species to the Earth's surface in the absence of precipitation, facilitated by gravitational settling, turbulent diffusion, or gas absorption (Sutton et al., 1995). As previously outlined, nitrate (NO$_3^-$_N) and ammonium (NH$_4^+$_N) constituted the dominant forms of nitrogen deposition in East Asian coastal wetlands, while the direct dry deposition of gaseous NOx and NH$_3$ contributed only 6.63 % and 9.73 % of total deposition, respectively. Building upon this, the present study further investigated the seasonal variation in nitrate and ammonium deposition, distinguishing between wet and dry fluxes across six representative regions (**Fig. 3**). Overall, nitrate deposition consistently surpasses ammonium deposition in all regions, reinforcing its dominant role in nitrogen input. Moreover, wet deposition was identified as the primary contributor across all areas, significantly exceeding dry deposition—underscoring the critical influence of seasonal rainfall on nitrogen dynamics in these ecosystems.

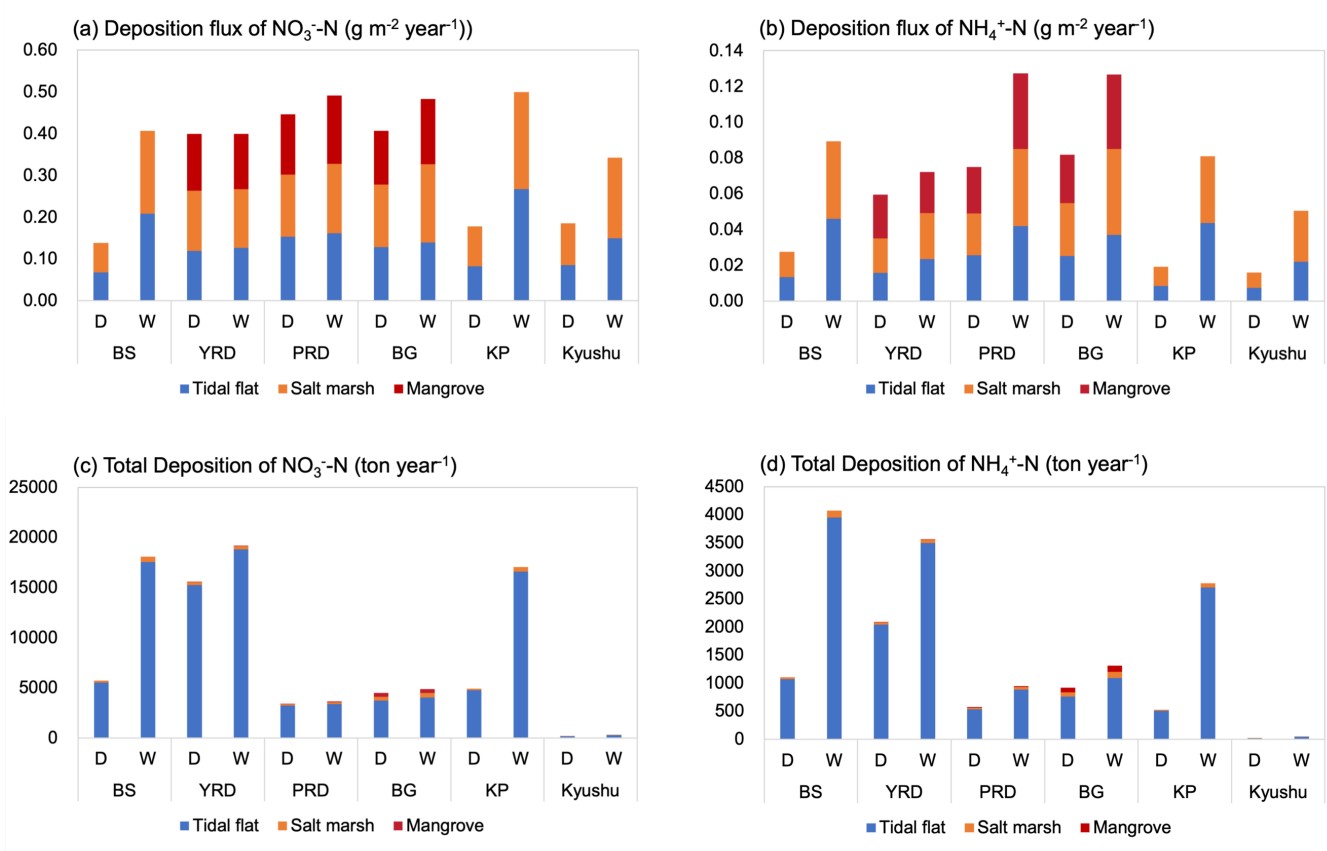

**Figure 3. Average dry and wet deposition fluxes of nitrate and ammonium nitrogen (a, b), and the total annual nitrogen deposition (c, d) in six typical wetland regions.**

However, the degree of difference between dry and wet deposition varies significantly among regions. For instance, the KP exhibited the most pronounced seasonal asymmetry in nitrate deposition, with wet deposition exceeding dry deposition by 0.17 g N·m$^{-2}$ yr$^{-1}$. A similarly marked difference was observed in the BS, where the nitrate wet-dry gap reaches 0.14 g N·m$^{-2}$ yr$^{-1}$, reflecting the combined influence of seasonal precipitation and emission intensity. The YRD distinguished itself with a relatively balanced level of ammonium nitrogen deposition, exhibiting a minimal dry-wet deposition difference of merely 0.05 g N m$^{-2}$ yr$^{-1}$, which was the smallest among all six evaluated regions.

In other words, even within regions that share similar latitudinal positions and ecological characteristics, the ratio of dry to wet deposition can exhibit significant variation due to differences in atmospheric circulation patterns, precipitation regimes, and land-sea interactions (Zhao et al., 2017; Zhang et al., 2006). For example, the annual dry and wet sedimentation fluxes of nitrate nitrogen and ammonium nitrogen in Kyushu of Japan are 183.92 tons, 337.05 tons, 15.45 tons, and 49.24 tons, respectively. Nitrogen deposition over Japan( Kyushu) is generally lower than that observed in heavily industrialized regions of China, a pattern supported by nationwide monitoring records showing comparatively modest wet and dry nitrogen inputs across Japan's coastal zones (Itahashi et al., 2021; Morino et al., 2011). These studies attribute the lower levels of nitrogen deposition largely to weaker local emission sources and the dominant influence of maritime air masses, which dilute atmospheric reactive nitrogen prior to deposition. This relatively low input of nitrogen is likely a consequence of limited industrial emissions coupled with the predominant influence of clean maritime air masses originating from upwind oceanic regions. (Hayashi et al., 2021; Kiriyama et al., 2021).

Focusing on specific types of wetlands, mangrove ecosystems consistently showed higher fluxes of both nitrate and ammonium nitrogen compared to salt marshes and tidal flats across most regions. This phenomenon may be attributed to their dense canopy structure, increased surface roughness, and complex vertical stratification, all of which significantly enhance nitrogen retention capacity. (Qian et al., 2023; Alongi, 2020). Furthermore, the proximity of numerous mangrove wetlands to urban and industrial sources likely contributes to elevated localized nitrogen loads. In contrast, tidal flats—often characterized by sparse vegetation—exhibit higher absolute fluxes of dry and wet deposition in some regions. This may be attributed to their increased

surface exposure and closeness to estuarine mixing zones, where atmospheric deposition readily interacts with open surfaces.

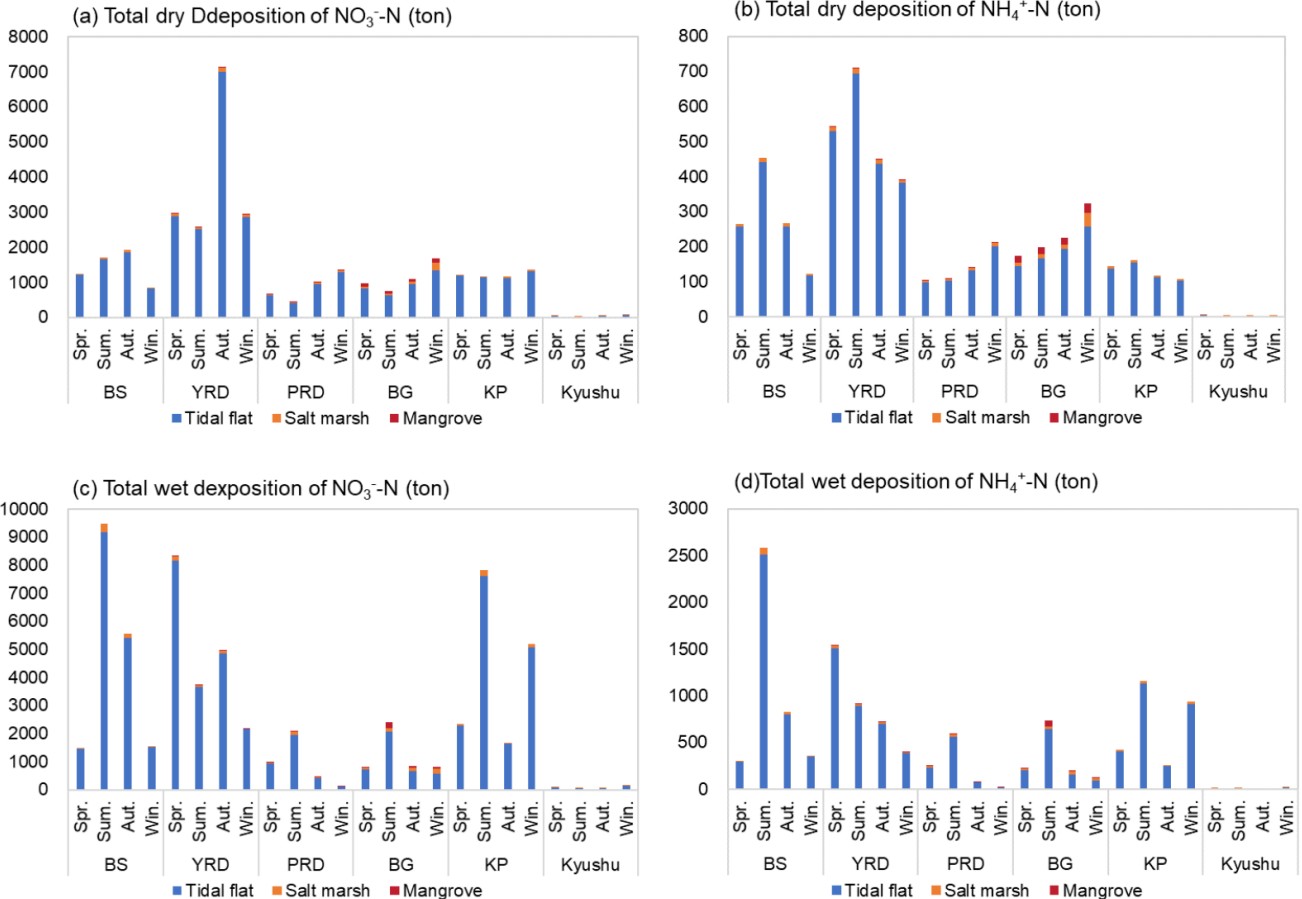

**Figure 4. Seasonal Characteristics of Total Nitrogen Deposition in Six Typical Regions. (a) and (c) show the total dry and wet deposition of nitrate nitrogen in different typical regions across different seasons; (b) and (d) show the total dry and wet deposition of ammonium nitrogen in different typical regions across different seasons.**

Seasonal patterns of total nitrogen deposition also vary substantially across coastal regions (**Fig. 4**). For instance, in the YRD, the dry deposition of nitrate nitrogen reached its peak during autumn, amounting to approximately 7139.55 tons. This enhancement is consistent with the seasonal intensification of fossil fuel combustion and industrial activity in eastern China, which maintains high $NO_x$ and oxidized nitrogen levels over the region (Sun et al., 2022). At the same time, meteorological conditions characteristic of autumn, including a shallower planetary boundary layer, more frequent stagnation, and reduced convective mixing, favour the accumulation and near-surface conversion of $NO_x$ to nitrate aerosols (Zang et al., 2022). In contrast, ammonium nitrogen dry deposition in the same region does not demonstrate a similar autumnal peak. This discrepancy is likely attributable to its stronger association with agricultural sources and their seasonally variable application cycles. (Kong et al., 2019; Liu et al., 2011).

In the BG, a distinct seasonal profile was observed. Here, dry deposition of both nitrate and ammonium nitrogen peaked in winter, which were 1667.85 tons and 322.32 tons respectively. This wintertime enhancement is consistent with the dominant influence of the East Asian winter monsoon, which transports $NO_x$ and $NH_3$-containing air masses from mainland China and Indochina toward the northern South China Sea and BG (Shah et al., 2020). This seasonal enhancement was likely driven by a combination of long-range pollutant transport via the East Asian winter monsoon, frequent atmospheric inversion events, and limited vertical mixing along the coast, all of which amplify nitrogen loading during the winter season (Lao et al., 2021).

Regarding wet deposition, a more consistent pattern was observed in the BS, YRD, PRD, and BG wetlands. In these regions, spring and summer seasons exhibited significantly higher deposition fluxes compared to autumn and winter. This seasonal increase aligns with enhanced convective activity and frequent rainfall events that facilitate the scavenging of soluble nitrogen compounds from the atmosphere (Tomczak et al., 2025). In contrast, Kyushu Island and the KP demonstrated relatively minor

seasonal variations in wet deposition fluxes for both nitrate and ammonium nitrogen, indicating a more uniform annual distribution. This smooth intra-annual profile was likely attributable to the maritime climate characteristic of these island regions, which features moderate temperature gradients and evenly distributed precipitation throughout the year (Zhao et al., 2015; Morino et al., 2011).

Overall, nitrogen deposition in East Asian coastal wetlands shows clear spatial and seasonal heterogeneity. Across all six regions, wet deposition contributes more to total nitrogen input than dry deposition for both nitrate and ammonium, indicating that precipitation plays a dominant role in transporting reactive nitrogen to coastal surfaces. Seasonal differences are particularly evident in spring and summer, when wet deposition fluxes markedly exceed those in autumn and winter. By contrast, dry deposition exhibits more region-specific behaviour; for example, the Yangtze River Delta shows a distinct autumn maximum in dry nitrate deposition, whereas the BG experiences its highest dry deposition in winter. These variations correspond closely to the differing meteorological conditions and emission intensities observed across regions and seasons. In areas such as Kyushu and the Korean Peninsula, seasonal differences in wet deposition remain relatively small, consistent with the more uniform precipitation and air-mass conditions characteristic of these maritime locations. Taken together, the results indicate that nitrogen deposition in coastal wetlands is shaped by both regional atmospheric environments and seasonal shifts in wet and dry deposition processes, with each region exhibiting a distinct seasonal signature.

**3.3 Potential Impacts of Atmospheric Nitrogen Deposition on Carbon Sequestration Capacity**

Utilizing remote sensing data and ecological model outputs from six representative coastal regions in East Asia, this study systematically evaluates the seasonal dynamics of NPP and $CO_2$ fixation across three wetland types (mangrove forests, salt marshes, and tidal flats) during spring, summer, autumn, and winter (**Fig. 5**). Among these wetland types, mangrove forests exhibited superior performance across all metrics, highlighting their crucial role as "blue carbon" ecosystems with significant carbon sequestration potential, consistent with previous global syntheses of mangrove carbon cycling (Alongi, 2014). During summer, mangrove forests in PRD received an average solar radiation of 1749.29 MJ $m^{-2}$ and APAR of 743.45 MJ $m^{-2}$, resulting in a high mean NPP of 776.16g C $m^{-2}$ approximately. Corresponding $CO_2$ fixation and $O_2$ release reached 2783.31 g $m^{-2}$ and 3244.35 g $m^{-2}$ respectively (Fig. 5a, Fig. 5b and Table S4), exceeding those in other regions. These values are on the same order of magnitude as those reported for highly productive mangrove stands in subtropical and tropical regions (Sun et al., 2024). FPAR for mangrove forests remained stable around 0.85 throughout the year, indicating a pronounced advantage in light use efficiency. This advantage arises from mangrove forests' evergreen habit, complex canopy structure, and higher water availability, which sustain efficient photosynthesis under high temperature and humidity conditions (Barr et al., 2013). Tidal flats, characterized by sparse or exposed vegetation cover, demonstrated the weakest carbon sequestration metrics. In winter, for example, APAR in tidal flats of the BS can decrease to 37.2 MJ $m^{-2}$, with NPP falling to 14.5 g C $m^{-2}$, and $CO_2$ fixation and $O_2$ release declining to 51.9 g $m^{-2}$ and 37.9 g $m^{-2}$, respectively. These low values align with the consistently low FPAR (<0.1), reflecting the limited photosynthetic contribution of tidal flats.

Seasonally, carbon sequestration exhibited peaks during the summer across all wetland types, primarily driven by maximal solar elevation angles, extended daylight hours, and increased radiation intensity. For example, the highest NPP of mangrove forests in summer reached 776.16 g C $m^{-2}$, approximately double that of winter values. This pronounced seasonal variation is even more pronounced in salt marshes and tidal flats, where NPP approaches near-zero levels during autumn and winter. Notably, in regions such as the PRD, mangrove NPP in autumn slightly exceeds spring values, likely attributable to nutrient inputs from post-typhoon events and heavy rainfall that temporarily enhance photosynthetic efficiency and biomass accumulation. (Qiu et al., 2019). Regional disparities in seasonal responses further underscore wetland ecosystems' sensitivity to localized climate and hydrological factors.

At the regional scale, a distinct north-to-south gradient in carbon sequestration capacity is evident (Fig. 5c). Overall, the PRD and YRD dominate total carbon production, contributing 3429.17 g C $m^{-2}$ and 3140.90 g C $m^{-2}$, respectively, reflecting their warm climates, high annual solar radiation, and long growing seasons. These southern regions supply substantial NPP flows to all three wetland types, particularly to mangrove forests, which receive 5384.90 g C $m^{-2}$ in total input—far exceeding

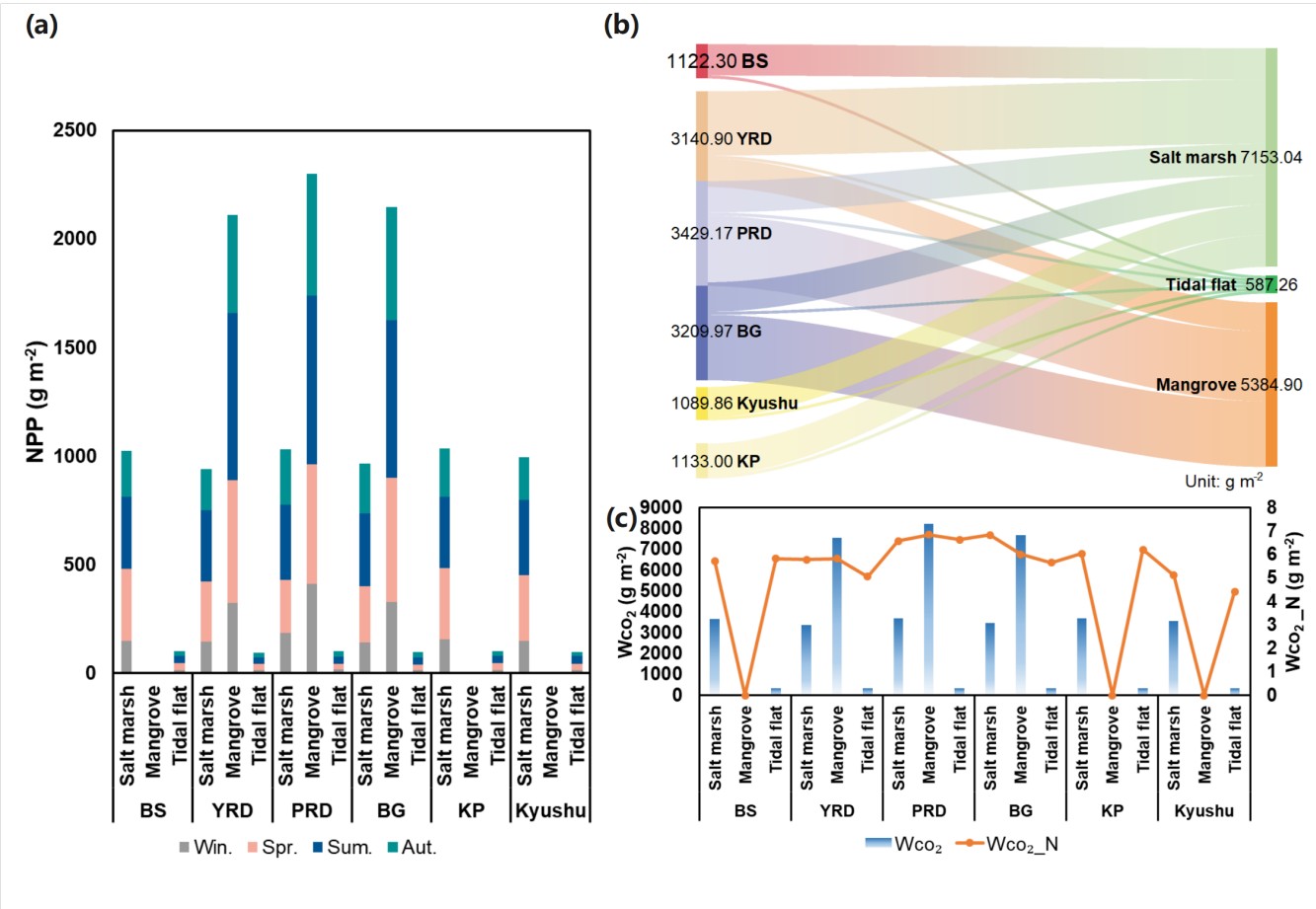

**Figure 5. NPP and carbon sequestration in different wetland types across various regions. (a) is the total NPP across four seasons for different wetland types in each region; (b) is the annual total carbon sequestration and nitrogen deposition-induced carbon sequestration in different wetland types across regions; the Sankey diagram shows the NPP distribution of the three types of wetlands in various regions (c).**

contributions from northern counterparts. In contrast, the BS and KP, located at higher latitudes with colder winters and lower annual irradiance, show markedly lower total NPP values (1122.30 and 1133.00 g C m⁻²), resulting in weaker connections to highly productive wetland types. The YRD and the PRD, located within tropical and southern subtropical climates, benefit from higher annual solar radiation and more stable light conditions (Alongi, 2014). These factors contribute to extended growing seasons and enhanced productivity of wetlands. For instance, mangrove forests in the region maintained the NPP exceeding 500 g C m⁻² during spring, summer, and autumn. In contrast, the NPP values of mangrove forests located in the YRD and BG regions were slightly lower. The limited winter solar radiation in these areas (total radiation < 745.00 MJ m⁻²), combined with a relatively high temperature threshold for salt marsh growth, significantly restricts photosynthetic activity. Conversely, mangrove forests sustained a winter NPP of 326.13 g C m⁻², demonstrating notable cold tolerance and stability as carbon sinks. Similarly, salt marshes and tidal flats within both the YRD and PRD exhibited elevated APAR and NPP values. This reinforces the conclusion that temperature and solar radiation serve as key external drivers limiting carbon sequestration in coastal wetlands.

The influence of nitrogen deposition on carbon sequestration also shows marked seasonal and regional variability (Table S4). The regulatory effect of nitrogen input on carbon sequestration is more pronounced during the warm season (April to October), when wetland vegetation exhibits heightened photosynthetic potential and nutrient uptake efficiency. For example, in PRD mangrove forests, nitrogen deposition increases carbon sequestration by up to 6.85 g m⁻² during summer, accounting for approximately 0.1 % of total seasonal carbon sequestration; however, this effect diminishes to below 0.06 % in winter. This seasonal disparity likely results from low winter temperatures limiting nitrogen mineralization and plant assimilation, thereby constraining nitrogen's regulatory role. Salt marshes display similar seasonal trends but with comparatively lower nitrogen responsiveness, indicating lower sensitivity to exogenous nutrient inputs than mangrove forests. Additionally, in regions with

high background nitrogen levels, such as the Yangtze River Delta, the marginal enhancement of carbon sequestration by nitrogen deposition tends to plateau, suggesting the onset of nitrogen saturation effects.(Lu et al., 2021).

**4. Conclusions**

This study utilized the WRF-CMAQ model, integrating multi-source emission inventories and high-resolution wetland land cover data, to comprehensively assess nitrogen deposition flux characteristics and their ecological impacts in East Asian coastal wetlands. The main findings are as follows:

Terrestrial anthropogenic emissions, primarily originating from industry, transportation, and agriculture, remain the
445 predominant contributors to nitrogen deposition. Deposition fluxes of particulate and gaseous nitrogen species from these sources are substantially higher than those associated with ship emissions. Nevertheless, ship emissions still make a notable contribution to particulate nitrogen deposition, accounting for 10.13 % of $NO_3^-$_N and 15.22% of $NH_4^+$_N fluxes. These contributions highlight the regional importance of maritime activities in the formation of nitrate and ammonium aerosols, particularly in coastal zones where port activity is intensive and the interplay between wet and dry deposition processes is
450 complex.

On a per-unit-area basis, salt marshes of the East Asia region coastal wetland exhibit the highest nitrogen deposition fluxes, significantly surpassing mangrove forests and tidal flats. However, due to their relatively limited spatial extent, mangrove forests receive less of the total regional nitrogen inputs compared to the more extensive salt marshes and tidal flats. This highlights the importance of considering both per-area flux and overall coverage when evaluating nitrogen deposition pressure.
Wetland-specific deposition patterns reveal that $NO_x$-N deposition is roughly 20 % higher in mangrove forests and tidal flats than in salt marshes, whereas $NH_3$-N deposition is more pronounced in salt marshes. The PRD, YRD, and BG emerge as nitrogen deposition hotspots, with annual fluxes exceeding 10 g N m$^{-2}$, markedly higher than levels observed in the KP and Kyushu coastal regions. The PRD, in particular, exhibits the highest nitrogen load due to a combination of dense industrial emissions and a globally significant port cluster, resulting in elevated atmospheric nitrogen driven by both terrestrial and
maritime sources, establishing a characteristic nitrogen deposition core in port-industrial complexes.

Marked seasonal and spatial disparities characterize nitrogen deposition dynamics. In the YRD, dry nitrate deposition peaks in autumn at approximately 7139.55 tons, driven by fossil fuel emissions and atmospheric stagnation, whereas wet deposition in the BS and BG can exceed 60 % of annual input during spring and summer. Wetland type also matters: mangrove wetlands receive up to twice the nitrogen fluxes of nearby salt marshes, due to higher canopy roughness and urban proximity.
Carbon sequestration exhibits clear spatial and temporal heterogeneity in response to atmospheric depositions, with southern wetlands displaying greater sink potential than northern counterparts. Mangrove forests in tropical regions sustain high NPP year-round (above 500 g C m$^{-2}$ in spring, summer, and autumn). Seasonal carbon sequestration reaches its maximum in summer as a result of higher solar radiation and longer daylight duration. Nitrogen deposition provides additional enhancement to carbon uptake during the warm season. It increases carbon sequestration by 6.85 g m$^{-2}$ in mangrove forests of the Pearl River
Delta, although this influence becomes much weaker in winter. In regions with high nitrogen load, signs of nitrogen saturation are evident, signalling the need for future nitrogen management strategies that account for both regional variability and spatiotemporal dynamics of nitrogen inputs.

This study establishes a connection between atmospheric nitrogen input into coastal wetlands and the natural carbon reservoir, thereby enhancing our understanding of nutrient sources and mechanisms within coastal ecosystems that play a crucial role in
mitigating climate change. Furthermore, these findings provide an alternative perspective on the potential impacts of nitrogen management stemming from anthropogenic activities.

**Code/Data availability**

The Final Analysis (FNL) meteorological data are available from the National Centres for Environmental Predictions (NCEP) at https://rda.ucar.edu/datasets/ds083.2. The base source code of CMAQv5.4 is available at https://github.com/USEPA/CMAQ.
The data of wetland types is available from GWL_FCS30 (https://doi.org/10.5194/essd-15-265-2023). Solar radiation data is

**Author contribution**

JL and YX analyzed the data; JL wrote the manuscript draft; SJ provided emission inventory data; YZ, YX, CJ, QY and WM reviewed and edited the manuscript.

**Competing interests**

The authors declare that they have no conflict of interest.

**Acknowledgement**

This work is supported by the Project funded by the National Natural Science Foundation of China (42375100), the Natural Science Foundation of Shanghai Committee of Science and Technology, China (22ZR1407700) and the 2024 AI for Science Project funded by Fudan University.

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
