# Peer review of "Atmospheric nitrogen deposition fluxes into coastal wetlands and their impacts on ecosystem carbon sequestration in East Asia"

_EGUsphere, 2025_

## Author Comment (AC1)

**Response**

We greatly appreciate the reviewer's thorough evaluation and constructive feedback. The comments have helped us significantly improve the scientific robustness and presentation of the manuscript. We have addressed each point in detail and revised the manuscript accordingly. A point-by-point response is provided below, with all modifications clearly indicated in the revised version.

**Reviewer #1**

This study applies a WRF–CMAQ modelling framework combined with multi-source emission inventories and high-resolution wetland maps to quantify nitrogen deposition in East Asian coastal wetlands and assess its impacts on carbon sequestration in different wetland types. The manuscript is logically organized and provides valuable model-based insights into source-specific nitrogen inputs and spatiotemporal deposition patterns. However, several methodological details, particularly the separation of ship and anthropogenic sources, the diagnosis of source-specific deposition fluxes, and the treatment of uncertainties, require clearer description before publication. At the same time, I think that the spatial distribution of nitrogen deposition should include the whole region, e.g. Yellow Sea, East China Sea. The manuscript has some originality and significance, but the writing is not rigorous enough, the arguments are not comprehensive enough, and the discussion is not indepth enough. Careful revision is recommended.

- 1. The number of references cited throughout the manuscript is insufficient, and many places that should be supported by citations are not. For example, in lines 215–230, many key concepts and mechanisms are presented without referencing the relevant literature.
- Thank you for your suggestion. We conducted a full-text review and added citations of references in some places where literature support was needed. The number of references has increased from 69 to 108 now.
- 2. The discussion is not sufficiently in-depth.
- Thank you for your suggestion. Additional content has been added to our Results and Discussion, such as comparing the simulation results of the impact of ship emissions of nitrogen deposition in this study with other studies (Section 3.1). The reasons for the seasonal differences in nitrogen deposition in different coastal wetland areas were further discussed and summarized (Section 3.2). The changes of NPP in coastal wetlands of East Asia were discussed from the perspectives of seasonal differences, regional differences and differences among different types of wetlands (Section 3.3). For specific modifications, please refer to Results and Discussion.

**• Section 3.1:**

[revised manuscript text omitted]

• Thank you for your suggestion. In Section 3.1, we added the contribution of ship emission inventories to other studies for comparison. Also, we have added comparisons with other studies in the discussion sections of Sections 3.1 and 3.2.

In summary, nitrogen deposition in East Asian coastal wetlands is shaped jointly by the type of emission sources and the chemical forms of nitrogen involved. Terrestrial anthropogenic emissions remain the primary contributor to total reactive nitrogen inputs, which is consistent with continental-scale assessments across China and other industrialized regions (Liu et al., 2024, 2013). However, ship emissions also exert an important influence on coastal atmospheric chemistry, particularly through the formation of secondary particulate nitrate and ammonium. The simulated contributions of shipping in this study align with previous findings that reported substantial maritime impacts on coastal nitrogen aerosols. Evidence from regional atmospheric modelling consistently indicates that NOx emissions of ships substantially elevate coastal nitrate concentrations, with increases of 20-30% reported for Chinese marginal seas (Lv et al., 2018), and analogous nitrate enhancements found in the Yangtze River Delta due to dense maritime traffic (Liu et al., 2016). Compared with these studies, our results reveal a comparable magnitude of shiprelated nitrogen deposition, particularly for particulate  $NO_3^--N$  and  $NH_4^+-N$  in areas with intensive port activity. This consistency reinforces the emerging understanding that maritime transport represents a significant and spatially focused source of nitrogen enrichment in coastal boundary layers, complementing the broader and more diffuse contributions from land-based anthropogenic sources.

4.In Figure 2, the N and E (north and east) indicators are missing from the map.

• The N and E (north and east) indicators are added in the Figure 2.

Figure 2. Spatial Distribution of annual total atmospheric nitrogen deposition in coastal wetlands across different regions of East Asia (g N  $m^{-2}$  yr $^{-1}$ ).

5.In Table 1, there is an extra horizontal line under "Anthropogenic".

• Thank you for your suggestion. The table seems to have extra line segments due to the issue of page spread. We have adjusted this table.

6.Does "total nitrogen deposition" refer only to inorganic nitrogen deposition, or to the sum of inorganic and organic nitrogen deposition?

• The relevant explanations have been added to the methodology.

In this study, total nitrogen deposition refers exclusively to total inorganic nitrogen (TIN), defined as the sum of oxidized inorganic nitrogen species ( $NO_2$ , NO,  $NO_3^-$ ) and reduced inorganic nitrogen species ( $NH_3$  and  $NH_4^+$ ).

7. Why choose 1, 4, 7, 10 to represent spring, summer, autumn, and winter instead of 12-2 to represent winter, and 3-5 to represent spring? Because this is not the result of a post-sampling experiment, so I think it's better to choose three months to represent a season.

On one hand, in this study we have to do a series of sensitive simulations for different emission categories in our experimental designs, the simulation of continuous months would require a lot of computational costs, which was not feasible in practical study. On the other hand, the selection of January, April, July and October to represent winter, spring, summer and autumn follows a widely adopted practice in regional atmospheric modeling studies that require seasonally representative simulations rather than continuous multi-month runs. These four months are commonly used as seasonal proxies because they correspond to the midpoint of each climatological season and minimize transitional effects associated with monsoon onset and withdrawal. Our goal was to capture the characteristic meteorological conditions, emission patterns and deposition behaviors of each season through independent simulations. January, April, July and October provide the most stable representation of seasonal atmospheric states while avoiding inter-month variability associated with early or late seasonal transitions. This approach is consistent with numerous WRF-CMAQ and regional climate studies that adopt single-month seasonal proxies when performing factorial or scenario-based experiments rather than long continuous runs. For this reason, we have added the following explanations of limitations in the methodology:

The use of a single representative month for each season is a methodological simplification relative to full three-month seasonal simulations. This choice was dictated by the factorial experimental design, which required independent simulations under two emission scenarios, and by the associated computational demands. Although this single-month representation is widely adopted in regional atmospheric modelling studies and has been demonstrated to capture the characteristic meteorological and chemical features of each season(Wu et al., 2021; Li et al., 2018), it inevitably introduces some degree of uncertainty related to intra-seasonal variability. Future work involving continuous multi-month simulations for each season would help further constrain this uncertainty.

8.I think the methods section does not clearly explain how different emission sources are distinguished. This may cause readers to question the robustness of your source-specific results.

• Thank you for your suggestion. We have added explanations on how to distinguish different emission source areas in the methodology. Specifically as follows:

In developing the emission inventories, this study explicitly separated terrestrial anthropogenic sources from marine ship emissions to enable a source-specific attribution of atmospheric nitrogen inputs to coastal wetlands. The land-based anthropogenic emissions for China were derived from the Multi-resolution Emission Inventory for China (MEIC), while emissions for other Asian regions were based on the MIX Asian inventory(Yue et al., 2017). Ship emissions were calculated using a bottom-up method based on AIS data with a fine resolution, following established practices for high-resolution marine emission modelling (Jiang et al., 2024; Fan et al., 2016). Based on the inventory of land-based anthropogenic, two parallel emission scenarios were constructed. The first scenario included both land-based anthropogenic and marine ship emissions (with shipping scenario). The second scenario excluded all ship emissions (without-shipping scenario). The contribution of shipping to nitrogen deposition was quantified by comparing the deposition fields simulated under these two scenarios.

9. The seasonal differences are not discussed in sufficient detail. For example, in lines 307–309: "In other words, even within regions that share similar latitudinal positions and ecological characteristics, the ratio of dry to wet deposition can exhibit significant variation due to differences in atmospheric circulation patterns, precipitation regimes, and land-sea interactions." You could try to incorporate backward trajectory analysis and precipitation data (if available) to support and deepen this discussion. There are also no relevant references cited here.

• We thank the reviewer for the insightful suggestion. We have now expanded the seasonal interpretation by incorporating mechanisms involving monsoon circulation, precipitation patterns, and marine air-mass transport, supported by relevant literature. Although backward-trajectory analyses were not conducted within the current study, we now cite previous trajectory-based studies over Kyushu and coastal Japan that demonstrate the influence of clean maritime inflow on nitrogen deposition patterns. This revision strengthens the mechanistic understanding of seasonal variability and improves the robustness of our interpretation. The specific modifications are as follows:

In other words, even within regions that share similar latitudinal positions and ecological characteristics, the ratio of dry to wet deposition can exhibit significant variation due to differences in atmospheric circulation patterns, precipitation regimes, and land-sea interactions (Zhao et al., 2017; Zhang et al., 2006). For example, the annual dry and wet sedimentation fluxes of nitrate nitrogen and ammonium nitrogen in Kyushu of Japan are 183.92 tons, 337.05 tons, 15.45 tons and 49.24 tons respectively. Nitrogen deposition over Kyushu is generally lower than that observed in heavily industrialized regions of China, a pattern supported by nationwide monitoring records showing comparatively modest wet and dry nitrogen inputs across Japan's coastal zones (Itahashi et al., 2021; Morino et al., 2011). These studies attribute the lower levels of nitrogen deposition largely to weaker local emission sources and the dominant influence of maritime air masses, which

dilute atmospheric reactive nitrogen prior to deposition. This relatively low input of nitrogen is likely a consequence of limited industrial emissions coupled with the predominant influence of clean maritime air masses originating from upwind oceanic regions. (Hayashi et al., 2021; Kiriyama et al., 2021).

10.it is better to use "Autumn" rather than "Fall."

 Thank you for your suggestion. The relevant expressions in the manuscript have been corrected in full.

[revised manuscript text omitted]

---

## Author Comment (AC2)

**Response**

We sincerely appreciate the reviewer thoughtful feedback and the time they dedicated to evaluating our work. Their comments have greatly helped us enhance the quality of our study and its presentation. We have carefully addressed each comment in the responses that follow and have implemented all changes in the revised manuscript.

**Reviewer #2**

This study used WRF-CMAQ with integrated high-resolution wetland type data, AIS-based ship emission inventories, and regional nitrogen deposition simulations to quantify nitrogen inputs to East Asian coastal wetlands from the perspective of source—sink coupling. The findings provide a scientific foundation for understanding how coastal ecosystems respond to anthropogenic activities and long-range nitrogen transport. This study has a certain level of innovation and logic. However, major revisions are still needed.

1.In this study, the simulated nitrogen deposition flux is the most important model result. However, there is no information about how the flux is simulated. I suggest adding the process in the Methods section.

• Thank you for your suggestion. We added the process of nitrogen deposition simulation to the methodology. Specifically as follows:

In this study, total nitrogen (TN) refers exclusively to total inorganic nitrogen (TIN), defined as the sum of oxidized inorganic nitrogen species (NO2, NO, HNO3/NO3-) and reduced inorganic nitrogen species (NH3 and NH4+). TIN was simulated for four representative months of 2017 (January, April, July and October), corresponding to winter, spring, summer and autumn. Selecting single representative months has been widely adopted in regional modelling to capture climatological seasonal characteristics under factorial experimental designs (Li et al., 2019; Qi et al., 2017). Each simulation used a five-day spin-up period to minimize the influence of initial conditions. In total, eight model experiments were conducted, consisting of four months and two emission scenarios. This simulation framework provided a consistent basis for evaluating both seasonal variations and the source-specific contributions of nitrogen deposition in East Asian coastal wetlands. The nitrogen deposition flux was directly output from the model. The dry deposition flux for each nitrogen species was calculated by the model based on the dry deposition velocity multiplied by the simulated surface-layer concentration. The wet deposition flux was simulated by scavenging nitrogen species from the atmosphere through both in-cloud and below-cloud processes.

2.The ship emissions inventory used in this study only considered NOx, NH3, PM2.5, and PM10. However, commonly used ship emissions inventories include SO2, NOx, PM2.5, CO, hydrocarbons, and GHG species (Yi et al., 2025). This study highlights the impact of nitrogen species on coastal wetlands; however, this limitation still needs to be mentioned, as these species can interact with each other.

• Thank you for the reviewers' suggestions. During the simulation process, the list used actually

did contain SO2, NOx, PM2.5, CO, hydrocarbons and types of greenhouse gases, but this was not expressed clearly in the previous manuscript. Therefore, we supplemented the relevant explanations in the manuscript and increased the citations of the literature. The added content in the manuscript is as follows:

The resulting ship emissions inventory includes nitrogen oxides  $(NO_x)$ , ammonia  $(NH_3)$ , and particulate matter  $(PM_{2.5}, PM_{10})$ , sulphur dioxide  $(SO_2)$ ,  $NO_x$ , carbon oxygen (CO), hydrocarbons, and greenhouse gas (GHG) species  $(Yi\ et\ al.,\ 2025)$ .

3.In the process of calculating carbon sequestration, the authors did not mention which parameters are based on model results and which are based on literature.

• Both Section 2.3 in the manuscript and Section 1.3 in the attachment elaborate on the calculation method of NPP. SOL is calculated from the Global High-Resolution (3-hourly, 10 km) Surface Solar Radiation Dataset (1983-2018, monthly) described in the attachment. The values of *FPAR* and ε are derived from the literature. All other parameters are derived from the model results. Relevant explanations have been supplemented in Section 2.3 of the manuscript. The added content in the manuscript is as follows:

In the CASA model, biome-specific constant FPAR values were assigned to different coastal wetland types to reflect their contrasting canopy structures and vegetation cover. Specifically, an FPAR of 0.85 was used for mangroves, consistent with satellite-derived APAR estimates for dense mangrove forests (Zheng and Takeuchi, 2022). A moderate FPAR of 0.65 was adopted for salt-marsh wetlands, in line with typical growing-season FPAR ( $\approx 0.4-0.7$ ) reported for marsh vegetation. For sparsely vegetated tidal flats, an FPAR value of 0.10 was chosen to represent the dominance of water and bare sediment and the low emergent leaf area during most tidal cycles (Hawman et al., 2023).

4.In Figure 1, the unit for nitrogen emissions is missing. Besides, the nitrogen emissions are not clearly defined: does the nitrogen here only include NO and NO2, or does it contain other species?

• Thank you for your suggestion. In the original figure, N refers to the nitrogen element. To reduce ambiguity, we have added the explanation of this part in our hands. The added content in the manuscript is as follows:

Overall, the nitrogen (N element) emission inventory and wetland type distribution in East Asia adopted in this study are shown in Fig. 1.

- 5. The first paragraph in Section 3.1 did not cite any figures, tables, or references. It is not clear where the results come from.
- Thank you for your suggestion. We added the citations of Table 1 and the references in the first paragraph of Section 3.1 of the manuscript.
- 6. There is a distinct mistake in Line 66 of the Supplementary Information (SI).

• Thank you for your suggestion. The reference error that existed here has been corrected.

**Reference**

- 1. Hawman, P. A., Mishra, D. R., and O'Connell, J. L.: Dynamic emergent leaf area in tidal wetlands: Implications for satellite-derived regional and global blue carbon estimates, Remote Sens. Environ., 290, 113553, https://doi.org/10.1016/j.rse.2023.113553, 2023.
- Li, J., Nagashima, T., Kong, L., Ge, B., Yamaji, K., Fu, J. S., Wang, X., Fan, Q., Itahashi, S., Lee, H.-J., Kim, C.-H., Lin, C.-Y., Zhang, M., Tao, Z., Kajino, M., Liao, H., Li, M., Woo, J.-H., Kurokawa, J., Wang, Z., Wu, Q., Akimoto, H., Carmichael, G. R., and Wang, Z.: Model evaluation and intercomparison of surface-level ozone and relevant species in East Asia in the context of MICS-Asia Phase III Part 1: Overview, Atmospheric Chem. Phys., 19, 12993–13015, https://doi.org/10.5194/acp-19-12993-2019, 2019.
- 3. Qi, L., Li, Q., Henze, D. K., Tseng, H.-L., and He, C.: Sources of springtime surface black carbon in the Arctic: an adjoint analysis for April 2008, Atmospheric Chem. Phys., 17, 9697–9716, https://doi.org/10.5194/acp-17-9697-2017, 2017.
- 4. Yi, W., Wang, X., He, T., Liu, H., Luo, Z., Lv, Z., and He, K.: The high-resolution global shipping emission inventory by the Shipping Emission Inventory Model (SEIM), Earth Syst. Sci. Data, 17, 277–292, https://doi.org/10.5194/essd-17-277-2025, 2025.
- 5. Zheng, Y. and Takeuchi, W.: Estimating mangrove forest gross primary production by quantifying environmental stressors in the coastal area, Sci. Rep., 12, 2238, https://doi.org/10.1038/s41598-022-06231-6, 2022.

---

## Author Response (AR1)

**Response**

We greatly appreciate the reviewer's thorough evaluation and constructive feedback. The comments have helped us significantly improve the scientific robustness and presentation of the manuscript. We have addressed each point in detail and revised the manuscript accordingly. A point-by-point response is provided below, with all modifications clearly indicated in the revised version.

**Reviewer #1**

This study applies a WRF–CMAQ modelling framework combined with multi-source emission inventories and high-resolution wetland maps to quantify nitrogen deposition in East Asian coastal wetlands and assess its impacts on carbon sequestration in different wetland types. The manuscript is logically organized and provides valuable model-based insights into source-specific nitrogen inputs and spatiotemporal deposition patterns. However, several methodological details, particularly the separation of ship and anthropogenic sources, the diagnosis of source-specific deposition fluxes, and the treatment of uncertainties, require clearer description before publication. At the same time, I think that the spatial distribution of nitrogen deposition should include the whole region, e.g. Yellow Sea, East China Sea. The manuscript has some originality and significance, but the writing is not rigorous enough, the arguments are not comprehensive enough, and the discussion is not in-depth enough. Careful revision is recommended.

1. The number of references cited throughout the manuscript is insufficient, and many places that should be supported by citations are not. For example, in lines 215–230, many key concepts and mechanisms are presented without referencing the relevant literature.

- Thank you for your suggestion. We conducted a full-text review and added citations of references in some places where literature support was needed. The number of references has increased from 69 to 107 now.

2. The discussion is not sufficiently in-depth.

- Thank you for your suggestion. Additional content has been added to our Results and Discussion, such as comparing the simulation results of the impact of ship emissions of nitrogen deposition in this study with other studies. The reasons for the seasonal differences in nitrogen deposition in different coastal wetland areas were further discussed and summarized. The changes of NPP in coastal wetlands of East Asia were discussed from the perspectives of seasonal differences, regional differences and differences among different types of wetlands. For specific modifications, please refer to **Results and Discussion**.

3. The manuscript lacks comparison with other related studies.

- Thank you for your suggestion. In Section 3.1(**Line 245-260, page 7**), we added the contribution of ship emission inventories to other studies for comparison. Also, we have added comparisons with other studies in the discussion sections of Sections 3.1 and 3.2 (**Line 275-295, page 9**).

[revised manuscript text omitted]

4.In Figure 2, the N and E (north and east) indicators are missing from the map.

- The N and E (north and east) indicators are added in Figure 2 (**Line 270-275, page 8**).

[Figure]

**Figure 2. Spatial Distribution of annual total atmospheric nitrogen deposition in coastal wetlands across different regions of East Asia (g N m⁻² yr⁻¹). BS means Bohai Bay, YRD means the Yangtze River Delta, PRD means Pearl River Delta, BG means Beibu Gulf, KP means the Korean Peninsula, and Kyushu means the Kyushu of Japan.**

80

5.In Table 1, there is an extra horizontal line under "Anthropogenic".

- Thank you for your suggestion. The table seems to have extra line segments due to the issue of page spread. We have adjusted this table (**Line 225-230, page 6**).

6.Does "total nitrogen deposition" refer only to inorganic nitrogen deposition, or to the sum of inorganic and organic nitrogen deposition?

- The relevant explanations have been added to the methodology (**Line 165-170, page 5**).

*In this study, total nitrogen deposition refers exclusively to total inorganic nitrogen (TIN), defined as the sum of oxidized inorganic nitrogen species ($NO_2$, $NO$, $NO_3^-$) and reduced inorganic nitrogen species ($NH_3$ and $NH_4^+$).*

7.Why choose 1, 4, 7, 10 to represent spring, summer, autumn, and winter instead of 12-2 to represent winter, and 3-5 to represent spring? Because this is not the result of a post-sampling experiment, so I think it's better to choose three months to represent a season.

- The selection of January, April, July and October to represent winter, spring, summer and autumn follows a widely adopted practice in regional atmospheric modeling studies that require seasonally representative simulations rather than continuous multi-month runs. These four months are commonly used as seasonal proxies because they correspond to the midpoint of each climatological season and minimize transitional effects associated with monsoon onset and withdrawal. In contrast, using multi-month periods such as December to February or March to May would require continuous model integration over several months, which was not feasible in this study due to computational cost and the experimental design. Our goal was to capture the characteristic meteorological conditions, emission patterns and deposition behaviors of each season through independent simulations. January, April, July and October provide the most stable representation of seasonal atmospheric states while avoiding inter-month variability associated with early or late seasonal transitions. This approach is consistent with numerous WRF-CMAQ and regional climate studies that adopt single-month seasonal proxies when performing factorial or scenario-based experiments rather than long continuous runs. For this reason, we have added the following explanations of limitations in the methodology (**Line 175-180, page 5**):

*The use of a single representative month for each season is a methodological simplification relative to full three-month seasonal simulations. This choice was dictated by the factorial experimental design, which required independent simulations under two emission scenarios, and by the associated computational demands. Although this single-month representation is widely adopted in regional atmospheric modelling studies and has been demonstrated to capture the characteristic meteorological and chemical features of each season(Wu et al., 2021; Li et al., 2018), it inevitably introduces some degree of uncertainty related to intra-seasonal variability. Future work involving continuous multi-month simulations for each season would help further constrain this uncertainty.*

8.I think the methods section does not clearly explain how different emission sources are distinguished. This may cause readers to question the robustness of your source-specific results.

- Thank you for your suggestion. We have added explanations on how to distinguish different emission source areas in the methodology. Specifically, as follows (**Line 155-165, page 4-5**):

*In developing the emission inventories, this study explicitly separated terrestrial anthropogenic sources from marine shipborne emissions to enable a source-specific attribution of atmospheric nitrogen inputs to coastal wetlands. The land-based anthropogenic emissions for China were derived from the Multi-resolution Emission Inventory for China (MEIC), while emissions for other Asian regions were based on the MIX Asian inventory(Yue et al., 2017). Shipborne emissions*

*were calculated using a bottom-up method based on AIS data, following established practices for high-resolution marine emission modelling (Jiang et al., 2024; Fan et al., 2016). Based on the baseline land-based anthropogenic inventory, two parallel emission scenarios were constructed. The first scenario included both land and marine emissions, hereafter referred to as the with-shipping scenario. The second scenario excluded all shipborne emissions, hereafter referred to as the without-shipping scenario. The contribution of shipping to nitrogen deposition was quantified by comparing the deposition fields simulated under these two scenarios.*

9. The seasonal differences are not discussed in sufficient detail. For example, in lines 307–309: "In other words, even within regions that share similar latitudinal positions and ecological characteristics, the ratio of dry to wet deposition can exhibit significant variation due to differences in atmospheric circulation patterns, precipitation regimes, and land-sea interactions." You could try to incorporate backward trajectory analysis and precipitation data (if available) to support and deepen this discussion. There are also no relevant references cited here.

- We thank the reviewer for the insightful suggestion. We have now expanded the seasonal interpretation by incorporating mechanisms involving monsoon circulation, precipitation patterns, and marine air-mass transport, supported by relevant literature. Although backward-trajectory analyses were not conducted within the current study, we now cite previous trajectory-based studies over Kyushu and coastal Japan that demonstrate the influence of clean maritime inflow on nitrogen deposition patterns. This revision strengthens the mechanistic understanding of seasonal variability and improves the robustness of our interpretation. The specific modifications are as follows (**Line 325-335, page 10**):

*In other words, even within regions that share similar latitudinal positions and ecological characteristics, the ratio of dry to wet deposition can exhibit significant variation due to differences in atmospheric circulation patterns, precipitation regimes, and land-sea interactions (Zhao et al., 2017; Zhang et al., 2006). For example, the annual dry and wet sedimentation fluxes of nitrate nitrogen and ammonium nitrogen in Kyushu of Japan are 183.92 tons, 337.05 tons, 15.45 tons, and 49.24 tons, respectively. Nitrogen deposition over Japan—including Kyushu—is generally lower than that observed in heavily industrialized regions of China, a pattern supported by nationwide monitoring records showing comparatively modest wet and dry nitrogen inputs across Japan's coastal zones (Itahashi et al., 2021; Morino et al., 2011). These studies attribute the lower levels of nitrogen deposition largely to weaker local emission sources and the dominant influence of maritime air masses, which dilute atmospheric reactive nitrogen prior to deposition. This relatively low input of nitrogen is likely a consequence of limited industrial emissions coupled with the predominant influence of clean maritime air masses originating from upwind oceanic regions. (Hayashi et al., 2021; Kiriyama et al., 2021).*

10. it is better to use "Autumn" rather than "Fall."

- Thank you for your suggestion. The relevant expressions in the manuscript have been corrected in full (**Line 345, page 11**).

[Figure]

**Figure 4. Seasonal Characteristics of Total Nitrogen Deposition in Six Typical Regions. (a) and (c) show the total dry and wet deposition of nitrate nitrogen in different typical regions across different seasons; (b) and (d) show the total dry and wet deposition of ammonium nitrogen in different typical regions across different seasons.**

160

**Reviewer #2**

This study used WRF-CMAQ with integrated high-resolution wetland type data, AIS-based ship emission inventories, and regional nitrogen deposition simulations to quantify nitrogen inputs to East Asian coastal wetlands from the perspective of source–sink coupling. The findings provide a scientific foundation for understanding how coastal ecosystems respond to anthropogenic activities and long-range nitrogen transport. This study has a certain level of innovation and logic. However, major revisions are still needed.

1. In this study, the simulated nitrogen deposition flux is the most important model result. However, there is no information about how the flux is simulated. I suggest adding the process in the Methods section.

- Thank you for your suggestion. We added the process of nitrogen deposition simulation to the methodology. Specifically, as follows (**Line 165-175, page 5**):

  *In this study, total nitrogen (TN) refers exclusively to total inorganic nitrogen (TIN), defined as the sum of oxidized inorganic nitrogen species ($NO_2$, $NO$, $NO_3^-$) and reduced inorganic nitrogen species ($NH_3$ and $NH_4^+$). TIN was simulated for four representative months of 2017 (January, April, July and October), corresponding to winter, spring, summer, and autumn. Selecting single representative months has been widely adopted in regional modelling to capture climatological seasonal characteristics under factorial experimental designs (Li et al., 2019; Qi et al., 2017). Each simulation used a five-day spin-up period to minimize the influence of initial conditions. In total, eight model experiments were conducted, consisting of four months and two emission scenarios. This simulation framework provided a consistent basis for evaluating both seasonal variations and the source-specific contributions of nitrogen deposition in East Asian coastal wetlands.*

2. The ship emissions inventory used in this study only considered $NO_x$, $NH_3$, $PM_{2.5}$, and $PM_{10}$. However, commonly used ship emissions inventories include $SO_2$, $NO_x$, $PM_{2.5}$, CO, hydrocarbons, and GHG species (Yi et al., 2025). This study highlights the impact of nitrogen species on coastal wetlands; however, this limitation still needs to be mentioned, as these species can interact with each other.

- Thank you for the reviewers' suggestions. In fact, during the simulation process, the list used did contain $SO_2$, $NO_x$, $PM_{2.5}$, CO, hydrocarbons, and types of greenhouse gases, but this was not expressed clearly in the manuscript. Therefore, we supplemented the relevant explanations in the manuscript and increased the citations of the literature. The added content in the manuscript is as follows (**Line 130, page 3**):

  *The resulting ship emissions inventory includes nitrogen oxides ($NO_x$), ammonia ($NH_3$), and particulate matter ($PM_{2.5}$, $PM_{10}$), sulphur dioxide ($SO_2$), $NO_x$, carbon monoxide (CO), hydrocarbons, and greenhouse gas (GHG) species (Yi et al., 2025).*

3. In the process of calculating carbon sequestration, the authors did not mention which parameters are based on model results and which are based on literature.

- Both Section 2.3 in the manuscript and Section 1.3 in the attachment elaborate on the calculation method of NPP. SOL is calculated from the Global High-Resolution (3-hourly, 10 km) Surface Solar Radiation Dataset (1983-2018, monthly) described in the attachment. The values of *FPAR* and $\varepsilon$ are derived from the literature. All other parameters are derived from the model results. Relevant explanations have been supplemented in Section 2.3 of the manuscript. The added content in the manuscript is as follows (**Line 210-215, page 6**):

  *In the CASA model, biome-specific constant FPAR values were assigned to different coastal wetland types to reflect their contrasting canopy structures and vegetation cover. Specifically, an FPAR of 0.85 was used for mangroves,*

*consistent with satellite-derived APAR estimates for dense mangrove forests (Zheng and Takeuchi, 2022). A moderate FPAR of 0.65 was adopted for salt-marsh wetlands, in line with typical growing-season FPAR (≈0.4–0.7) reported for marsh vegetation. For sparsely vegetated tidal flats, an FPAR value of 0.10 was chosen to represent the dominance of water and bare sediment and the low emergent leaf area during most tidal cycles (Hawman et al., 2023).*

4.In Figure 1, the unit for nitrogen emissions is missing. Besides, the nitrogen emissions are not clearly defined: does the nitrogen here only include NO and $NO_2$, or does it contain other species?

- Thank you for your suggestion. In the original figure, N refers to the nitrogen element. To reduce ambiguity, we have added the explanation of this part in our hands. The added content in the manuscript is as follows (**Line 140, page 4**):

*Overall, the nitrogen (N element) emission inventory and wetland type distribution in East Asia adopted in this study are shown in Fig. 1.*

5.The first paragraph in Section 3.1 did not cite any figures, tables, or references. It is not clear where the results come from.

- Thank you for your suggestion. We added the citations of Table 1 and the references in the first paragraph of Section 3.1 of the manuscript (**Line 220-225, page 6**).

6. There is a distinct mistake in Line 66 of the Supplementary Information (SI).

- Thank you for your suggestion. The reference error that existed here has been corrected (SI).

**Reference**

[revised manuscript text omitted]